# Exploring Elementary Students’ Social-Emotional Development Through Dialogic Pedagogy: Insights from Cinematic Narratives

**DOI:** 10.3390/bs15121701

**Published:** 2025-12-08

**Authors:** Fatma Aslantürk Altıntuğ, Ahmet Güneyli

**Affiliations:** Faculty of Education, European University of Lefke, 99728 Lefke, Cyprus; faslanturk@eul.edu.tr

**Keywords:** social and emotional learning (SEL), dialogic pedagogy, film-based learning, bullying in schools, critical thinking

## Abstract

Scholars of social and emotional learning (SEL) emphasize conversation as a valuable pedagogical strategy; however, they often provide limited guidance on how such discussions can be effectively implemented in classroom practice. This study explores the potential of film-based dialogic pedagogy as a means to foster students’ social and emotional learning by engaging them in reflective dialogues and collective meaning-making. The participants were primary school students from Northern Cyprus. Adopting a qualitative case study design, the research observed students’ emotional and behavioural responses during and after a film screening. Multiple data sources—focus group discussions, individual interviews, unstructured classroom observations, and written reflections—were analysed to gain a comprehensive understanding of students’ experiences. Findings illustrate how dialogic engagement through film stimulated emotional awareness and social sensitivity, particularly toward issues of bullying. The study indicates that film-based dialogic pedagogy can enrich students’ emotional and cognitive engagement with social issues, promote critical reflection on diverse perspectives, and cultivate a sense of community grounded in respect, empathy, and reciprocal participation.

## 1. Introduction

The study delves into the transformative potential of dialogic pedagogy in modern education, a pedagogical approach that places a premium on student discourse and collaborative knowledge construction to foster critical participation as well as exploring the development of social and emotional learning (SEL). Recently, many studies have conducted studies to explore the ways through which dialogic pedagogy can be implemented ([4]; [19]; [41]; [49]). This study uses dialogic pedagogy as an approach to empower students social and emotional learning SEL which is essential as it helps students to develop vital skills that support both academic success and personal growth. As a war and contact zone, Cyprus has always seen surges of refugees and migrants, and over the past 20 years, it has been a popular destination for international students. Pluralistic pedagogies adorned with nationally monoglossic discourses have become iconic pressing trends in education in the age of unprecedented diversity and emphasised internationalist theories worldwide, causing a dichotomous doom-loop with regard to multifaceted parameters of learning environments ([70]). The high flow of migration leads to various cultural and socio-economic tensions within the education system, which increase bullying cases among students at schools ([59]).

Bullying is such an important issue that it can have a negative impact not only on a person’s school life but on their whole life. The person being bullied can experience traumatic effects and emotions. It is therefore extremely important to learn how to deal with bullying and how to prevent it. Studies show that ([24]; [89]) employing approaches and pedagogies that do trigger student voice and create discursive places in a class, help reducing bullying. Dialogic pedagogy is one of the approaches that help teachers create discursive places in a class. In their study ([7]) the researchers investigated the impact of implementing dialogic gatherings in two elementary classrooms, focusing on their potential to prevent school violence. The findings of their study revealed that dialogic gatherings successfully raised students’ understanding of the difference between violent and non-violent interactions, as well as encouraging social cohesion and non-violent behaviours. Likewise, in his research [74] ([74]) utilised dialogic events to analyse bullying and interpersonal disputes. The study’s findings revealed that valuing students’ event-rich conversations and dialogues can help them receive insightful advice and resolve disputes more skilfully. These articles offer insightful information about how to use dialogic pedagogy to handle and stop bullying in classrooms.

Even though the research on Social and Emotional learning (SEL) and dialogic pedagogy stresses the value of empathy, inclusive classroom environments, and conversation, there are still a number of significant gaps. First off, prior research on SEL that used films or multimodal narratives frequently focused on emotional engagement—the way that media arouse empathy or emotion—rather than talking about how these feelings are changed through dialogic interaction ([79]; [26]). This “affect-induction bias” runs the risk of turning films become emotional triggers instead of interactive educational resources. Second, the utilisation of real classroom discussion data has a methodological flaw. Few studies evaluate actual conversation transcripts that show how students absorb meaning, emotion, and moral reasoning in real time, whereas the majority of study papers now in existence focus on survey or reflection-based outcomes ([39]). Third, because the majority of earlier research was done in Western, monolingual contexts, its contextual reach is still somewhat small. In multicultural and post-conflict environments like Northern Cyprus, where social tension and diversity issues are embedded in everyday school life, little is known about how dialogue-based, film-based Social and Cultural Interaction (SCI) practices function ([70]). In order to overcome these drawbacks, the current study explores how film-based dialogic pedagogy might help primary school pupils in a culturally diverse Cypriot school develop social and emotional intelligence skills like empathy, self-awareness, and social awareness. This study provides qualitative insight into how students’ emotional reactions to the movie Wonder become dialogic resources for moral reasoning, peer empathy, and group meaning-making by examining their written and spoken reactions.

This study’s main goal is to investigate the transformative potential of dialogic pedagogy as a means of improving social and emotional learning (SEL) and dealing with bullying in educational environments that are culturally diverse. The study specifically explores how, in a setting characterised by migration and intercultural tensions, like Cyprus, dialogic, student-centred methods might promote social cohesion, emotional intelligence, and critical participation among students. This study provides a unique viewpoint by directly connecting dialogic pedagogy to SEL development and anti-bullying tactics in a culturally and socioeconomically complex setting, even though previous research has looked at the theoretical foundations of dialogic pedagogy and its application in classrooms. A distinct geopolitical and cultural component is added to the research by the Cypriot environment, which is defined by its status as a war and contact zone as well as its growing migrant population. This focus widens the present debate on dialogic education beyond mainstream Western contexts and engages with underexplored multicultural settings.

This study is significant because it addresses pressing issues in education in classrooms that are becoming more and more diverse. Although cultural diversity brought about by migration has been enriching, it has also created tensions that show up in educational settings as an increase in bullying and exclusion. There is an urgent need for educational practices that assist students’ personal and interpersonal development in addition to academic accomplishment, as bullying has long-lasting emotional and psychological repercussions, particularly on young learners. By fostering inclusive, discursive environments, dialogic pedagogy becomes a relevant and effective strategy for fostering empathy, voice, and community in the classroom.

This study makes several important contributions to educational research and practice. It challenges the dominance of nationally monoglossic educational discourses and embraces the complexities of culturally diverse classrooms by highlighting the applicability and relevance of dialogic pedagogy in educational setting in Cyprus. By providing a conceptual framework that links student voice, emotional growth, and social responsibility, the study theoretically improves the connection between dialogic pedagogy and social and emotional learning (SEL). Practically speaking, it offers empirical evidence of how dialogic classroom activities—like dialogic meetings and discussions that are full of events—can successfully lessen bullying, encourage non-violent conduct, and cultivate respect and understanding between students from different backgrounds. Furthermore, the results have significant policy ramifications, indicating that dialogic pedagogy can be an effective part of teacher education programs and SEL curriculum that aim to establish inclusive, courteous, and emotionally supportive learning environments.

### Research Case and Questions

The main reason for conducting the study in the context of Northern Cyprus is the unique socio-political and cultural structure of the region, which is defined as a “war and contact zone” ([86]). Northern Cyprus is a region characterized by high rates of immigration, increasing ethnic and cultural diversity and the resulting socio-economic tensions in the education system. This has led to an increase in bullying cases in schools ([77]). Therefore, this study is meaningful in terms of investigating the applicability of inclusive approaches such as dialogic pedagogy and SEL in this unique geographical context, which is at the intersection of cultural diversity and social conflict. This focus widens the current debate on dialogic pedagogy beyond mainstream Western contexts and engages with underexplored multicultural settings.

Social and Emotional Learning (SEL) enables individuals to develop basic skills such as empathy, self-awareness, building relationships, and managing their emotions ([33]). Dialogic pedagogy aims to support students’ social and emotional development in an environment of dialogue and collaboration with active participation ([32]). The study focused on bullying because the “discursive spaces” provided by dialogic pedagogy allow students to express their feelings and thoughts, understand the perspectives of others, and learn a culture of mutual respect and support within the community. This environment offers a proactive method for combating bullying ([74]). Dialogic pedagogy becomes a relevant and effective strategy for fostering empathy, voice, and community in the classroom.

Although many problems originating from students are observed in primary schools in Northern Cyprus, bullying is one of the most devastating effects among these problems ([29]; [57]). Therefore, it is of great importance to examine the phenomenon of bullying in an academic context by separating it from other problems. It causes devastating results not only at the individual level but also at the level of the classroom and school climate. The main reasons why bullying is distinguished from other student-related problems are its prevalence, power of impact and long-term consequences. Developmental psychology and educational sciences literature show that bullying experiences experienced in primary school can spread to the entire life of the individual. The systematic nature of bullying negatively affects not only the victims but also the observers and the bullies. This makes bullying a problem that needs to be solved both at the individual and institutional level. Bullying has a wider impact than other student-related problems such as discipline problems, failure in classes, and lack of attention. Because bullying directly threatens the social fabric of the school. A student cannot be expected to feel safe, be academically successful, or establish healthy social relationships in an environment where they are bullied. Therefore, it can be said that bullying plays a role as one of the underlying causes of other problems ([11]; [62]; [67]; [81]).

The reason for choosing elementary school students for the study is that this age group is in a critical developmental phase where the foundations of social-emotional skills are being laid. SEL skills developed at an early age enable children to establish healthier peer relationships, increase their emotional awareness, and develop strategies for coping with bullying ([2]; [88]). In addition, methods based on creative and emotional interactions, such as film-based learning and dialogic pedagogy, can be effective teaching tools at a time when abstract thinking is just beginning to develop in this age group. Social, emotional, mental, physical, and moral qualities are typically thought of as the cornerstones of all learning and competencies in childhood.

This study is thought to make three main contributions in terms of expanding the SEL and dialogic pedagogy literature. i. Cultural context diversity: Conducted in an understudied, multicultural, and migration-shaped educational context such as Northern Cyprus, this research goes beyond Western-cantered studies and reveals how dialogic pedagogy functions in different geographical-sociological conditions. ii. Integration of SEL with film-based teaching: By showing how media tools such as film can be integrated into social-emotional teaching, the impact of cinema as a new pedagogical tool is emphasized. iii. Data from the student perspective: By analysing children’s personal experiences with bullying and their perceptions of teacher, peer, and family roles, it is shown how SEL is shaped from the student perspective. In summary, it is thought that this research will allow universal inferences to be made for both teacher education and school policies with the micro-analysis method centred on the student perspective. Thus, the study reveals that dialogic pedagogy is not only a pedagogical but also a sociocultural transformation tool. Research questions of the study are given below:How do students’ views and feelings shape their understanding of peers’ and teachers’ responses to bullying after film-based dialogic pedagogy?How does film-based dialogic pedagogy change students’ views on teachers’ roles in addressing bullying?How does dialogic film discussion affect how students see peers’ responses to bullying?After film-based dialogic pedagogy, how do students view families’ role in helping bullied children?

## 2. Theoretical Framework: Dialogic Pedagogy and Social and Emotional Learning

Over the past two decades, there has been a burgeoning interest in pedagogical strategies that amplify student voice and engagement, marking a departure from monologic instruction that stifles communicative inclusivity and collaboration ([75]). Dialogic pedagogy, as defined by [75] ([75]), champions a teaching and learning milieu where both students and teachers engage actively in addressing genuine problems, thereby cultivating a multitude of perspectives, critical examination of knowledge claims, and a community ethos predicated on respect, support, and reciprocal participation norms.

This educational approach not only facilitates expressive student participation but also engenders an interactive, collaborative learning environment that transcends conventional curricular confines to encompass societal issues, thereby enriching students’ cultural and social identities as well as developing social and emotional learning through dialogue ([32]). The contrast between monologic and dialogic discourse, as explored by [60] ([60]), highlights the significance of dialogic instruction in stimulating student thought and fostering an environment where students, rather than passively receiving information, actively contribute to their own knowledge construction through dialogic engagement.

Prominent scholars, including [83] ([83]), [5] ([5]) and [3] ([3]), have contributed to the conceptualization and empirical validation of dialogic pedagogy. [83] ([83]) proposed a model of enquiry-oriented curriculum that emphasizes the co-construction of knowledge through teacher-student dialogues, while [5] ([5]) and [3] ([3]) illustrated the critical role of ‘scaffolded dialogue’ in enabling students to express their ideas and engage with diverse perspectives, thereby fostering a ‘pedagogy of mutuality’ that recognizes students as competent contributors to their learning process.

This approach not only enhances critical and analytic thinking skills but also cultivates cultural competence and critical consciousness, empowering students to challenge societal norms and contribute to social transformation ([34]; [37]) and the development of social and emotional learning ([32]).

In essence, dialogic pedagogy emerges as a pivotal educational paradigm that not only promotes student agency, voice, and collaborative learning but also prepares students for active, critical, and democratic citizenship. By embracing dialogic pedagogies, educators can create inclusive classroom communities where diverse cultural and social experiences are valued and leveraged as assets for learning and critical engagement with the world.

Dialogic pedagogy is presented as an innovative educational approach that values and prioritizes student voice, collaborative knowledge construction, and the negotiation of meaning within the classroom. This approach is rooted in the belief that education should cultivate democratic communicative norms and active citizenship, thereby enabling students to engage meaningfully in societal discussions and controversial issues ([75]; [43]; [3]; [63]).

The pedagogy is characterized by its emphasis on authentic problem-solving, where both students and teachers play active roles in the joint construction of knowledge, fostering a classroom culture that is supportive, inclusive, and based on mutual respect ([75]). It encourages students to express their diverse identities and perspectives, thus promoting a deep understanding and respect for cultural and social differences.

Dialogic pedagogy contrasts sharply with traditional classroom discourse, which often silences student voice and inhibits active participation. In response, educational theorists and practitioners have developed various dialogic approaches, such as Accountable Talk, Exploratory Talk, Collaborative Reasoning, and Dialogic Teaching, to recenter student dialogue in the learning process ([55]; [54]; [66]; [3]).

[60] ([60]) and [83] ([83]) explore the nuances of dialogic instruction, highlighting the importance of dialogue in creating a classroom environment where students are not passive recipients of knowledge but active contributors to their own learning journey. These interactions are not only educational but also formative, as they contribute to students’ personal, social and emotional development.

[5]’s ([5]) and [3]’s ([3]) work further illuminates the concept of scaffolded dialogue and dialogic teaching, underscoring the transformative power of talk in the educational process. He identifies talk as the foundational element of learning, facilitating children’s development of identity, self, and worth through engaging with the diverse voices within their cultural and historical contexts. Alexander’s principles of dialogic teaching emphasize collective, reciprocal, supportive, cumulative, and purposeful classroom discourse, aimed at fostering an environment where students can freely express their thoughts and ideas ([3]).

Through these dialogic practices, students are encouraged to engage in critical thinking, reasoning, and justifying their viewpoints, thereby cultivating a classroom atmosphere that is inclusive and conducive to the sharing of cultural and social narratives. This shift towards dialogic pedagogy represents a significant move away from traditional teaching models, offering a pathway to more engaged, inclusive, and democratic forms of education which contribute to the development of social and emotional learning as well.

Students improve their social-emotional capabilities as well as their critical thinking and communication abilities when dialogic pedagogy is incorporated into classroom environments ([20]). Dialogic pedagogy and social and emotional learning (SEL) are intrinsically related because they both stress the value of empathy, active listening, teamwork, and self-awareness ([39]). Students gain important SEL competencies through meaningful communication, including how to negotiate different points of view, constructively express their emotions, and have polite conversations ([50]). Dialogic pedagogy fosters a classroom atmosphere that enhances social awareness, emotional control, and responsible decision-making through scaffolded discourse and reciprocal engagement. In their case study ([36]) used Dialogic Literary Gatherings to encourage students to communicate their emotions, think critically about their sentiments, and have sympathetic conversations. The dialogic atmosphere also promoted a sense of respect and community among the students. Additionally, dialogic pedagogy builds students’ emotional resilience and interpersonal skills by encouraging a sense of belonging and respect for one another ([36]). This equips students to participate as socially conscious, introspective, and sympathetic people both within and outside of the classroom.

Like other educational trends, the idea of SEL and its promotion arose in a particular historical and cultural setting, and it is important to recognise that it is a component of a larger therapeutic shift in education ([13]; [27]; [73]). For the time being, it is particularly crucial to address two assumptions: (1) cultural beliefs about the emotions that influence school SEL conversations, and (2) educational beliefs about the best way to conduct SEL discussions.

Social and Emotional Learning (SEL) instruction is of great interest to educators, policymakers, and researchers as a way to help children develop their social and emotional abilities and create a healthy school climate ([9]; [25]). Research has demonstrated that Social and Emotional learning (SEL) enhances students’ mental health, mental achievement, and civic engagement while also lowering risk behaviours, anxiety, and depressive symptoms ([47]; [23]; [42]).

The Collaborative on Academic, Social, and Emotional Learning ([15]) established five interconnected competency categories that make up Social and Emotional Learning (SEL): self-awareness, self-management, social awareness, interpersonal skills, and responsible decision-making. According to research, successful SEL implementation in schools is linked to better academic achievement, more student engagement, better social skills, and a decrease in disruptive classroom behaviours, suspensions, and other forms of disciplinary action ([16]).

Assessments of the students revealed improvements in a range of SEL competencies. They showed social awareness by relating to classmates and characters, and self-awareness by identifying and expressing their own feelings. When students talked about managing their anger or reacting coolly to taunts, it was clear that they were practicing self-management. Through group discussions, listening to various viewpoints, and supporting their peers, they improved their interpersonal skills. Lastly, their recommendations for handling bullying situations showed that they could make responsible decisions by considering the repercussions and acting in a way that was sympathetic and constructive to society.

Therefore, development of SEL is crucial as it gives students the capacity to control their emotions, monitor their behaviour, build and sustain healthy relationships, and manage their learning. Employing new discursive programmes and pedagogies is necessary to advance SEL in classrooms ([28]; [47]). It has been suggested that having classroom discussions is a key strategy for accomplishing SEL goals ([87]). SEL can be implemented through direct instruction or indirect instruction ([21]; [39]). As the screens cover an important place in children’s lives, media can be used as a tool to develop SEL ([25]). In order to support and enhance the active learning process of students, technology should be used to help them build their knowledge and skills while taking into consideration their age, developmental stage, and unique cognitive and emotional traits ([48]). Through the camera, kids are able to see things that were otherwise nearly impossible to view ([1]). Using films to teach social and emotional skills is an appealing idea because they can be an engaging medium to get kids interested in learning activities ([26]). According to several studies ([78]; [69]; [31]; [90]), film can convey social contexts and emotions more effectively than written texts or literature, supporting the aforementioned claim.

At the same time, films are effective mediums for moral and emotional instruction. They have the power to emotionally captivate audiences, cultivate empathy, and stimulate critical thinking regarding moral quandaries, interpersonal relationships, and social conventions ([17]; [53]; [65]). Compared to traditional storytelling methods, the use of music, speech, and pictures immerses viewers in the story and increases the effect of moral and emotional messages. According to this study, dialogic pedagogy offers the interactional framework that activates Social and Emotional (SEL) competencies, while the movie acts as an emotional and narrative catalyst in the dialogic learning process. By using particular dialogic “moves” that correspond with the five SEL subdomains that [15] ([15]) defined, the mechanisms that connect these domains can be comprehended. When students express how they feel about certain movie sequences and think about how their experiences and the characters’ experiences are similar, self-awareness begins to develop. As students express coping mechanisms and control their emotions during group conversations, self-management skills grow. Students’ social awareness is enhanced when dialogic inquiry promotes perspective-taking; they think about how others could feel or behave differently in comparable circumstances. While responsible decision-making skills grow as students assess the moral quandaries portrayed in the movie and jointly negotiate just, socially useful solutions, relationship skills are developed through collaborative reasoning, turn-taking, and empathic listening during peer debate. In this sense, film-based dialogic pedagogy serves as a cohesive framework wherein films evoke feelings, conversations turn those feelings into ideas, and ideas in turn strengthen social-emotional growth ([3]; [32]; [39]).

## 3. Method

### 3.1. Research Model

This study employs a case study design, a methodological approach within qualitative research paradigms ([85]). Throughout the research process, students’ emotional and behavioural responses during and after the film screening were observed, and comprehensive data were gathered through individual and group discussions. An unstructured observation form was utilized during the film screening, and the data collection process was further enhanced through individual interviews, focus group discussions, and the analysis of written documents. The study aims to examine how students’ coping mechanisms for school bullying, as well as their empathy, emotional awareness, and critical thinking skills, are influenced and developed through the medium of film.

### 3.2. Participants

The participants of the study comprised students selected from a primary school located in Northern Cyprus. The purposive sampling method was employed for participant selection. The researchers chose to work with a school that was most accessible, cost-effective, and time-efficient for them, resulting in the inclusion of a total of 35 students in the study (See Table 1). While accessibility was considered, this school was chosen because its students reflect the increasing ethnic, linguistic, and socio-economic diversity in Northern Cyprus. This diversity, which includes many migrant students and those with special educational needs, can lead to higher risks of bullying and social-emotional challenges. As a result, the school offers an important setting to study how film-based dialogic teaching might encourage empathy, inclusion, and better peer relationships. The students participated in focus group discussions organized into groups of seven and contributed data through individual interviews and written documents. Participation in the study was voluntary, with consent obtained from both the parents of the participants and the school administration. The participants’ personal information was treated with strict confidentiality.

The class comprises 35 students, including 18 girls and 17 boys. Of these, 17 are enrolled in Year 4 and 18 in Year 5. The cohort consists of 15 Cypriot, 10 Turkish, 5 Kurdish, 3 Russian, and 2 Arab students. Twenty students are immigrants, while fifteen are non-immigrants. Four students have special needs: two with autism spectrum disorder and two with attention deficit hyperactivity disorder.

### 3.3. The Role of Researchers

The researchers of this study, who facilitated and observed the data collection, have extensive experience in working with children in schools and are trained in dialogic and inclusive teaching methods. One of the researchers is a male professor who has been in the field of teacher education for 20 years. His subject areas are elementary pedagogy, multicultural education, special needs, native language teaching, and curriculum. The other is a female assistant professor with 25 years of teaching experience in educational sciences, specializing in dialogic pedagogy, inclusive education, and foreign language teaching. Their backgrounds helped ensure that interviews and group discussions with children were ethical and sensitive to children’s emotional well-being, cultural diversity, and issues such as bullying.

Two researchers played active roles in the research process. One researcher facilitated the dialogue processes, while the other was responsible for note-taking, observation, and recording. The researchers designed an unstructured observation form and conducted their observations using this tool during the data collection process. Furthermore, in both individual and focus group interviews, they refrained from directing the students’ responses, instead adopting an open-ended and interpretative approach that allowed the students to freely express their feelings and thoughts. After the interviews, the researchers provided mutual feedback on the process and integrated their perspectives to ensure consistency in the data analysis.

### 3.4. Data Collection Tools

Several data gathering techniques are used to investigate the moral teachings and themes presented in Stephen Chbosky’s Wonder movie. This enables researchers to look at how the movie may be used as a lens to study topics like diversity, bullying, identity formation, and resilience in educational settings. Stephen Chbosky’s film “Wonder” is full with moral lessons that can be used as a tool to explore certain issues. Wonder is a 2017 production. It is a purposeful and educationally sound option for a text. Bullying and diversity in multicultural classrooms are real-life situations that are reflected in the film’s realistic portrayal of a youngster with facial differences struggling with inclusion, exclusion, and empathy in a school environment. Elementary school students can identify, express, and critically discuss feelings like guilt, fear, compassion, and courage—all essential elements of SEL—through the film’s emotionally charged yet developmentally appropriate story ([15]; [25]). Wonder serves as a multimodal “shared text” that promotes group interpretation and moral reasoning in the context of dialogic pedagogy ([3]; [32]). Students can place themselves in relationships with a variety of characters, including Auggie, their friends, and their professors, through viewing and discussion. This helps them build empathy and perspective-taking abilities, which are essential for dialogic interaction and SEL objectives. Furthermore, Wonder depicts complex social connections (friendship, exclusion, resilience, and family dynamics) that encourage more in-depth, deliberate discussion and the collaborative creation of meaning, in contrast to other children’s films that oversimplify moral conflict. Because of its proven ability to elicit dialogic, emotional, and ethical interaction, the film was selected not just for its relevance to bullying but also as an effective teaching tool for promoting social-emotional development in diverse classrooms. Therefore, it has recently been used as a lens by scholars to look into the way that differences and disabilities are portrayed in the media, to explore initiatives to combat bullying and promote inclusion as well as investigating identity formation and resilience ([6]). The film “Wonder” is significant because it inspires in addition to entertaining. It instils in us a number of moral principles, including perseverance, culture, struggle, socioeconomic distinctions, love, and the power of familial love. Viewers are prompted to adopt a kinder, more compassionate perspective by its poignant depiction of a little boy’s quest for acceptance in a world that is ready to pass judgement. The movie provides educators and community leaders with a solid starting point for conversations about diversity, resiliency, and what beauty really means. Wonder’s message to “choose kind” has a profound effect that goes well beyond the movie theatre. Movies have many purposes in society than just providing after-work entertainment. On the other hand, it can also serve as a persuasive tool for educational information ([80]). A good film is one that uses text, sound, and visuals to depict society as it actually is ([46]). Consequently, the audience’s thoughts can be shaped by films in line with the filmmaker’s intention to communicate the message’s content.

The SEL framework is thus embodied in the real classroom narrative, and its theme findings (students’ emotional reactions, perceptions of staff, classmates, and families) represent dialogic teaching in action. It offers a lively, developmentally appropriate setting where children may identify, express, and assess feelings like empathy, guilt, fear, and fortitude. Wonder portrays complex interpersonal relationships involving peers, teachers, and families, in contrast to many children’s films that present simplistic moral dichotomies. This makes Wonder highly aligned with the principles of dialogic pedagogy, which value multiple perspectives, moral reasoning, and collaborative meaning-making ([3]; [83]; [32]). As a result, the film offers students a guided, emotionally engaging setting in which to practice dialogic thinking and SEL competencies within a realistic narrative context.

Although the study’s conclusions are limited in their generalisability due to its reliance on a single cinematic text, this design decision is consistent with a qualitative case study methodology that values interpretation depth over breadth. Long-term emotional engagement, dialogic inquiry, and in-depth theme analysis of student responses were made possible by concentrating on a single story; these outcomes can be undermined by watching several films. Future research, however, might broaden this strategy by include more films or multimodal texts to compare dialogic and affective results across narrative kinds.

The following tools were utilized to collect data in the study:i.*Unstructured Observation Form:* Developed by the researchers, this form comprises five main sections: ‘General Observation Information,’ ‘Emotional Reactions,’ ‘Behavioural Reactions,’ ‘Evaluation of the Environment,’ and ‘Free Observation Notes of the Researchers.’ During the film screening, students’ feelings, thoughts, and experiences related to the film were observed, and data were gathered using the unstructured observation form. The section titled ‘Observation Information’ included details such as the date, location, duration, and the names of the observers. The observation criteria are provided in Appendix B.ii.*Individual Interview Questions:* Students were asked to express their feelings and thoughts about the film through open-ended and interpretative questions.iii.*Focus Group Interview Questions:* These were designed to allow students to share their thoughts about the film through group interaction. Audio recordings were made to capture the dialogues and interviews in detail.iv.*Written Documents:* Students were provided with questions about the film and asked to respond to them in writing.

The same set of questions was used in the individual and group interviews as well as the written questionnaire. The forms consisted of a total of 8 open-ended questions, some of which included one or two sub-questions. The question form is provided in Appendix C. In developing the interview questions, the research questions were used as a basis, and expert feedback was obtained from one educational sociologist and one educational psychologist to assess the clarity, comprehensibility, and relevance of the questions to the scope of the study. Additionally, a pilot study was conducted with three students who were not part of the research group.

### 3.5. Process

First Day: The film Wonder was screened in the hall of the university where the researchers worked. Before the film, a preliminary interview was conducted with the students on the title and visuals of the film on the basis of dialogic pedagogy. During the film viewing process, students’ emotional and behavioural reactions were observed using an unstructured observation form. Two researchers reported students’ feelings and behaviours before and after watching the film based on the unstructured observation form; then the reports of the two researchers were integrated. Similar and different opinions in the reports were categorised. Similar opinions were written as a whole text. The different opinions were read, disagreements or differences were resolved, consistency was ensured and the issues that the two researchers agreed on were added under the report.

Second Day: The day after the film screening, the researchers went to the students’ schools and conducted individual interviews. One-to-one dialogues were established with the students using open-ended questions focused on interpretation and critical thinking. While one researcher guided the questions, the other took notes of important points and key words. After the one-to-one interviews, the students were randomly divided into groups of 3–5 students and focus group interviews were conducted. In the one-on-one interviews, main questions were asked to the students, while probe questions were used in the focus group interviews. In the one-on-one and focus group interviews, the researchers conducted interviews with the students in accordance with the dialogic pedagogy. This process was both designed as a teaching approach and recorded for data collection. Several structured processes were used to encourage dialogic participation: (1) Framing questions: To promote different viewpoints, open-ended questions like “How do you think Auggie felt in this scene?” or “Have you experienced something similar before?” were used at the start of each exercise. (2) Dialogical scaffolding: To preserve interactional flow, researchers demonstrated polite turn-taking, paraphrasing, and probing strategies (e.g., “Can you elaborate more on why you think that?”). (3) Thinking circles: After watching the movie or reading the parts, students gathered in small groups to discuss their ideas. The facilitator made sure that those who were more reserved had a chance to speak and that their answers built upon those of their peers. (4) Collaborative synthesis: To emphasise consensus, variety, and moral awareness, developing ideas were summarised on the board during a brief class discussion at the end of each session. By transforming the classroom into a dialogical space where students created meaning around Wonder’s moral and emotional quandaries, these facilitative strategies linked common conversations to the five SEL competencies: self-awareness, self-management, social awareness, relationship skills, and responsible decision-making ([15]).

To make sure the students felt secure and at ease during the interviews, a number of ethical and instructional strategies were implemented. Before gathering data, the students participated in a quick warm-up and introduction. To reduce any power disparity, the researcher sat at eye level with them. While avoiding directive language, open-ended questions were tailored to the children’s developmental stage and included illustrative examples where needed. Children were made aware that they could end the interview at any moment and cease answering questions. This created a secure and welcoming interview setting that encouraged kids to express themselves honestly.

Third Day: Finally, questions about the film were distributed to the students and they were asked to return with written answers one day later. Thus, the process of obtaining data was formed by obtaining individual dialogues, group dialogues, written documents and observation reports. SEL-based dialogic pedagogy practices, which also provided data collection, were implemented in six stages. This process is summarized and presented in Table 2.

### 3.6. Data Analysis

The data obtained from observations and interviews were combined as a single text. The collected data were analysed by content analysis method. The process consists of the following steps:Coding the Data: Observation forms, individual and group interview records, written documents were analysed and the data were coded thematically.Determination of Themes: Main themes and sub-themes were formed by bringing the codes together.Integration of Opinions: The reports prepared by the two researchers were compared, similar views were brought together and consensus was reached on different views.Ensuring Consistency: The final report was prepared in line with the consensus reached between the researchers.Verification: The audio recordings and written notes were reviewed and checked for consistency. Detailed analysis process is given in Appendix A.

In order to ensure validity in the study, data collection tools were designed with a multi-method approach; observation, individual interviews, focus group interviews and written documents were used together. Interview questions were developed in line with the opinions of an educational sociologist and an educational psychologist who are field experts and were tested with a pilot application. In order to increase the reliability of the data, two researchers were included in the research process; one conducted the observation and note-taking task, while the other facilitated the dialogue processes. The data obtained after the observation and interview were compared, similar opinions were combined, and differences were integrated through consensus. The audio recordings and written notes were reviewed again and subjected to content analysis, and thus internal consistency was achieved in the analysis process.

During the data analysis process, the two researchers first identified a question to be analysed. Within the scope of this study, the first interview question related to the first research question (How did his teachers behave and what did they do when Auggie was made fun of and excluded?) was determined. The content analysis for the first question was conducted separately by two researchers, and codes (keywords) were determined from each participant student’s responses to the first question. The consistency of the codes determined by the researchers was then examined. Based on the content analysis of the first question, the two researchers used [56] ([56]) coefficient of reliability for the codes they identified. [56] ([56]) suggest that there should be at least 70% consistency between coders. After examining the data generated by the examination of the first interview question, the researchers in this study arrived at a total of 24 codes. Nineteen of these codes were determined to be common codes, in other words, to be identical. In this case, the reliability coefficient between the two researchers was found to be 79% (19/24 × 100). After identifying the codes, the two researchers worked together each time, combining the codes to determine the themes. The same approach was adopted for the content analysis of other questions, observation data, and student assignments. Each time, the two researchers worked together, eliminating any differences in analysis to ensure consistency between the coders. Clusters of comparable meanings were gathered and classifications produced through open coding were compared to create themes. After the coders established a consensus, each theme was improved, and its scope was examined in accordance with the codebook.

To increase the study’s trustworthiness, a number of reliability techniques were used. Following the interviews, the results were first given to the instructors, and member verification was done to make sure the children’s explanations were appropriately represented. The thick description approach was used to provide a detailed report on the research context, participant characteristics, and interview findings. During the data interpretation process, researchers engaged in comparison talks to guarantee consensus among themselves; including several research viewpoints improved the themes’ robustness.

## 4. Findings

The findings of this study are based on the code book presented in Appendix A.

*Research Question 1*: Participants’ personal perspectives and emotional reactions to how a student experiencing bullying at school is approached. This research question yielded four themes from the data:Sadness: The most common reaction among students facing bullying was sadness. They expressed feeling very bad when mocked, often stating that the experience emotionally shook them. Some students described this through crying or a desire to be alone.Inner Struggle: Some students reported that bullying pushed them to make more friends or impacted their self-confidence. These students attempted to rebuild themselves internally while trying to display resilience against the bullying.Outward Response: A portion of the students took proactive steps to address the bullying. These actions included reporting the bullying to their teachers or directly confronting the bullies. These students adopted a more active approach to resolving the situation rather than staying silent.Seeking Connection and Support: Some students preferred to seek support from family or friends when faced with bullying. This search for support aimed to alleviate the effects of bullying. In particular, family support played a crucial role in helping them cope with the impact of bullying.

These themes are summarized in Table 3 below and are connected to the students’ perspectives as participants.

Teachers should be both empathetic supporters and disciplinarians, according to many students. “The teacher did not yell when she stopped the fight; instead, she told us to think about how the student being bullied felt,” said one participant. I was upset by that and thought about it before laughing once more. This exemplifies the self-awareness and social awareness skills of SEL that are acquired via introspective discussion as opposed to punishment. “If the teacher just punishes, the bully will get angrier; but if he speaks, everyone listens,” said another youngster. These quotations support [3]’s ([3]) theory of reciprocity pedagogy and show how dialogic exchanges turn disciplined times into cooperative moral reasoning.

*Research Question 2:* According to students’ perspectives, what are the roles and responsibilities of their teachers regarding bullying in school? As a result of the analysis regarding this research question, four themes emerged, which are briefly explained below (See Table 3):i.Discipline: The students participating in the study expressed how their teachers ensured discipline in response to negative behaviours such as bullying in school. Actions like punishing, warning, or taking students to the principal were frequently mentioned. Discipline was the most commonly expressed theme, with participants emphasizing how teachers should implement discipline in the face of negative behaviour.ii.Empathy: Participants frequently mentioned the importance of teachers approaching students in difficult situations with empathy. Empathy refers to understanding and trying to grasp the emotions of students in challenging circumstances, like Auggie. The study revealed that some teachers view empathy as a tool for resolving issues.iii.Support: This theme relates to protecting and supporting students who are in difficult situations. Participants noted that teachers stand by students who are bullied or excluded, defending them and providing emotional support.iv.Intervention: Efforts to prevent disputes or fights among students were grouped under this theme. Participants highlighted the role of teachers in stopping conflicts and calming students down. (See Table 4).

Teachers were acknowledged by students as important sources of both emotional support and discipline. Some stressed corrective action, reflecting students’ expectations of accountability and fairness: “The teacher should warn them … and punish them until they stop” (S33). Others appreciated compassion and empathy: “Teachers should treat them well … and understand them” (S5), “The teacher should ask other students, ‘What if you were in their shoes?’” (S12)—indicating knowledge of social awareness and emotional attunement in SEL ([15]). Statements like “The teacher should spend time with them” (S19) and “Protect them from being bullied” (S16), which are compatible with relationship skills and responsible decision-making, also highlighted supportive responsibilities. When seen through the lens of dialogic pedagogy ([3]), these observations show that kids view teachers as both agents of empathy and authority figures, demonstrating how polite conversation can promote moral and emotional learning.

*Research Question 3:* According to students’ opinions, how do peers approach students who experience bullying at school? When the opinions related to this question were analysed, peers’ approaches were divided into two categories: positive and negative (See Table 5).

Students reported both positive and negative peer responses. “They go to the park and have fun with him” (S18) and “They might tell people off” (S33) are two examples of highlighted prosocial behaviours that exhibit social responsibility, empathy, and inclusivity and correlate to social awareness, relationship skills, and responsible decision-making within the SEL framework ([15]). Others acknowledged unfavourable reactions that showed a lack of self-control and the maintenance of aggressive norms, such as “They might hit people who exclude him” (S27) or “They might not even look at you” (S24). These divergent opinions were turned into teaching opportunities through dialogic discussion following the watching of Wonder. Students thought of different, prosocial ways to respond and demonstrated how dialogic education ([3]) turns ordinary discourse into moral and emotional insight.

Positive approaches include forming friendships, providing support, and seeking help.

Forming friendships: This theme includes the views of peers who get along well with the bullied student, play with them, and spend time together. Participating students shared examples of peers who treat the bullied student well, avoiding exclusion.Providing support: Under the theme of support, participating students mentioned that peers display positive behaviours such as standing up for and protecting the bullied student. Efforts to empathize and boost the morale of a friend in distress were also expressed under this theme. Warning the bully was another behaviour of peers identified as supportive.Seeking help: Participating students suggested that peers might ask teachers for help to protect the bullied student.

Negative approaches were presented in two themes:Punishing: Some students stated that peers might try to support the bullied student by punishing or using violence against them.Bullying: Some participating students pointed out that, rather than supporting or helping, peers might engage in bullying themselves or ignore the bullied individuals altogether.

*Research Question 4.* According to student views, how families approach students who experience bullying at school (See Table 6).

Students stressed how important family support was to survivors’ emotional healing. “They should try to be very nice to him” (S33) and “I like that my family always supports me” (S4) are two examples of how many people valued warmth and trust, highlighting the family’s role in developing relationship skills and self-awareness within the SEL framework ([15]). Others identified empathy and understanding as essential traits: “It was nice that they made him feel like a special child and showed understanding” (S11), which reflects the SEL domains of empathy and social awareness. However, other students criticised inconsistent or controlling family behaviour, pointing out conflicts between autonomy and care: “Sometimes they show love but don’t respect their child’s decisions” (S14). These sentiments illustrate how dialogic discussions encourage students to critically evaluate emotional support and boundaries at home, transforming personal observations into moral reasoning consistent with dialogic pedagogy ([3]).

Positive Behaviour: Most participants emphasised that Augie’s family was loving and supportive towards their children. In particular, they stated that family members always defended their children who were bullied or had special needs and that they trusted them. Empathy and understanding were also the most prominent positive characteristics of the families.

Negative Behaviours: Some participants revealed oppressive attitudes and unbalanced relationships within the family. In particular, unequal treatment between siblings was criticised. In particular, the fact that the sister was given less attention and some of her decisions were not respected was characterised as negative.

Mixed Feelings: Some participants expressed that although the family was generally well-intentioned, they did not fully understand special children like Augie or respect their decisions enough.

## 5. Discussion

The explanations offered here are placed within the institutional and cultural framework of a Northern Cyprus primary school. As a result, rather than being general conclusions, the results should be viewed as context-sensitive patterns. The study does not assert causality or generalizable effects, but it does provide insights into potential ways that film-based dialogic teaching may improve Social and Cultural Development abilities.

*The study’s participant students’ feelings and opinions about bullying in schools:* The research’s findings, which highlight the participants’ feelings and opinions about bullying in schools, demonstrate the multifaceted emotional impacts of bullying on students. It is believed that psychological resilience, social support, and the capacity to handle bullying all influence students’ responses in dialogic pedagogy techniques. According to this research, depressive and lonely feelings are among the most frequent consequences of bullying; yet, some students said that they attempted to address the problem by re-establishing their confidence or making new acquaintances. Conversely, some kids have made an effort to deal with the issue head-on by addressing the bullies or their teachers. Research indicates that children who actively seek out social support—particularly from their families—are able to mitigate the negative impacts of bullying. These findings highlight the value of emotional support systems in the battle against bullying in schools once more. According to [58] ([58]), addressing students’ emotional needs and creating a safe haven for them at the school when they experience bullying can lessen the impact of horrific bullying events. Having social support networks around oneself can be quite helpful in dealing with bullying. Through the improvement of empathy and social skills, research ([45]; [52]) has demonstrated the effectiveness of teacher assistance and personal development activities in decreasing bullying in schools. Furthermore, it has been found that the relationship between the student and the teacher is the most essential component, and that students’ intrinsic motivation to stop bullying is increased when this relationship is supportive—that is, when the relationship is positive.

*Students’ opinions of how teachers handled bullying in the study:* The study participants anticipate that teachers will respond to bullying in schools by enforcing discipline. The literature views teachers’ roles in upholding discipline as essential to fostering a supportive school environment. Discipline is a particularly useful tool in the fight against bullying, according to models like the Olweus Bullying Prevention Program ([61]). The literature does, however, regularly note that this strategy is insufficient on its own and that discipline need to be reconstructive and educational as opposed to punishing ([22]; [84]).

Teachers’ empathy for students who have been bullied is another crucial quality that pupils look for in a teacher. The students in this study anticipate that teachers will act appropriately, try to understand the sentiments of the bullied kids, and approach them with empathy. This result is consistent with a large number of academic studies. Specifically, empathy is suggested by the Restorative Approach as a useful strategy to stop bullying and help victims reintegrate into society ([82]). Additionally, empathy can also play an educational role for bullying students. Empathy building activities for students who show bullying behaviour can reduce the likelihood of bullying recurrence.

One of the key responsibilities teachers have in the school environment is to assist students who are being bullied. The study’s participants expect teachers to defend individuals who are bullied and provide them with emotional support. According to the research, teachers’ supportive roles are essential to students’ social and emotional growth. For instance, a study conducted by [18] ([18]) reveals that students who are victims of bullying can recover faster with teacher support. At the same time, such support can prevent the recurrence of bullying and increase the self-confidence of victimised students. The students who participated in the research expect teachers to intervene directly in arguments and fights and to play a calming role. Researchers such as [76] ([76]) state that with an effective intervention, teachers can prevent the growth of conflicts between students and produce long-term solutions. To summarise, this study revealed that bullying in schools is perceived as a serious problem requiring adult intervention. As [68] ([68]) emphasised, it is seen that the help of bullied students from teachers and adults is an effective strategy in solving bullying incidents in schools.

*The views of the students in the research on the approaches of peers to bullying:* The study’s participants’ opinions on their bullied classmates fell into two broad categories, based on the research findings: positive approaches and negative approaches. Building relationships, offering assistance, and asking for aid were classified as positive ways. This method suggests that children who have been bullied anticipate social and emotional support from their classmates, and that assistance can take on both emotional and physical forms.

Treating bullied peers well, playing games and spending time with them was often mentioned. This result suggests that it is critical to shield students who are being bullied from peer rejection. The literature suggests that peer support is important in reducing the social isolation of victims of bullying and increasing their self-esteem ([12]). Friendship plays an important role in the emotional well-being of victims of bullying and is effective in strengthening social bonds ([44]; [72]). The study’s participants stressed that peers demonstrated actions like standing up for their friends who were being bullied, offering moral support, and showing empathy for them. One of the most crucial emotional defences that peers cultivate against bullying is empathy. Empathy is considered an important social skill in combating bullying and contributes to the emotional recovery of bullying victims ([35]). Such acts of support from students foster social cohesion and lessen the detrimental impacts of bullying.

Peer-related negative circumstances involving bullying were described as punishment and the peers themselves bullying the victim. Some students develop aggressive and damaging methods and resort to punishment in order to protect their bullied friends. The approach of punishing bullies shows that the methods used to combat bullying are sometimes misunderstood and can lead to further violence. In the literature, it has been stated that violence and punitive approaches can continue the bullying cycle and make the problem more complicated instead of solving it ([30]).

On the other hand, some students stated that they excluded or ignored their bullied peers rather than supporting them. This finding shows that bullying is closely related to social exclusion and loneliness. Bullying behaviours are often associated with power imbalance in the social hierarchy and exclusion is an important factor that reinforces this power imbalance ([71]).

*The views of the students in the study on the approaches of families to bullying:* In the research, students emphasised that it is important for families to show love and support to their children who are bullied. Children can feel more powerful when their families support them and believe in them. Family support has been shown in the research to be crucial for bullying victims’ emotional healing ([51]). Parents’ comfort and support of their children in particular make children feel safer when they are subjected to bullying. Families’ empathy and understanding are crucial in helping children who have been bullied emotionally recover. Families with empathy are better equipped to comprehend their children’s emotions and provide them with the support they need. According to research, children from sympathetic homes are better able to handle bullying ([10]). Anticipating empathy and comprehension from their relatives can aid in making youngsters feel understood, particularly those with special needs.

Some students in the study said that their family behaved in an oppressive and imbalanced way. The literature emphasises that children who experience oppressive and authoritarian parental attitudes may develop social issues and a lack of confidence ([8]). Some participants, on the other hand, stated that families generally have good intentions but they do not fully understand their children or do not respect their decisions sufficiently. This expresses the belief that parents are unable to provide their children with the whole support system they need to combat bullying.

These results highlight the close connection between social and emotional learning (SEL) and dialogic pedagogy. In addition to expressing their experiences and emotions, students also acquire critical SEL competencies like empathy, emotional control, social awareness, and responsible decision-making through dialogic classroom practices like open discussions, group reflections, and shared meaning-making. The findings portray that students can interact with emotionally complex characters and real-world problems in a safe, supervised setting by seeing films like Wonder, which provide rich stimulus for these discussions. In addition to this, students can discuss different viewpoints, respectfully express themselves, and develop stronger interpersonal relationships through this dialogic engagement which enables them to gain the required skills that are crucial for reducing bullying and creating inclusive, emotionally intelligent classroom environments.

Even though the study revealed recurring themes of empathy, moral reasoning, and introspection among participants, variations within subgroups might have affected how students perceived inclusivity and bullying. For instance, during conversations, boys tended to concentrate on justice and teacher authority, whereas girls tended to emphasise friendship and emotional understanding. These subtle differences, albeit not thoroughly examined, imply that kids’ engagement with dialogic and SEL-focused tasks may be influenced by their gender, age, and individual experiences with bullying. In order to effectively contextualise students’ moral and emotional growth, future studies should investigate similar tendencies using bigger, more demographically diverse samples.

## 6. Conclusions

The results of this study have a lot in common with the fundamental ideas of dialogic pedagogy. In particular, students’ expression of their own feelings (sadness, internal conflict, seeking support) and their efforts to develop solutions in their reactions to bullying situations allow their voices to be heard and their subjective perspectives to be included in the learning process. Students’ expectation of empathy, support and intervention from teachers positions the teacher not only as a transmitter of information but also as a dialogue partner who accompanies the student’s emotional and social process. The fact that students consider their friends’ attitudes from both positive (support, friendship, empathy) and negative (bullying, exclusion) aspects points to the importance of pluralism and different perspectives in learning environments. In this context, the study is notable for the mutual production of meaning and the development of critical thinking within the community that dialogic pedagogy aims for. The findings also provide rich data on the core areas of social emotional learning (SEL): self-awareness, self-management, social awareness, relationship building skills, and responsible decision making. Living in a safe and supportive environment is undoubtedly very important for the child’s development. However, a child who is used to having problems solved by the teacher and family may not be able to solve bullying or other problems on their own. Continually relying on outside assistance might not be able to stop bullying. It is crucial for the children to enhance their own abilities and acquire problem-solving skills. As a matter of fact, in this study, internal struggle was revealed by children as a solution suggestion. Feeling sad is a natural emotional reaction, but the fact that it was the most common finding in this study suggests that children are often helpless in the face of bullying. Finally, confronting the bully directly is a courageous behaviour, but there is also a risk of increasing the severity of conflicts. It may be useful to support the problem-solving skills of students experiencing bullying and to provide them with social skills training. Imaginary conversations and dialogues can enhance the experiences of students who have not experienced bullying or have only experienced it in small doses. Feelings such as sadness, internal conflicts, and loneliness experienced by students who are exposed to bullying provide a strong foundation for the development of self-awareness and emotional recognition skills. At the same time, some students’ behaviours, such as appealing to the teacher or confronting bullying, indicate self-management and problem-solving skills. Situations such as being supported by friends, being empathized with, or being excluded are critical in terms of social awareness and managing relationships. The empathy, support, and sense of trust that students expect from teachers and family figures also align with SEL’s goal of creating safe social environments. Therefore, the findings of the study demonstrate why SEL-based programs are necessary in schools. It is essential that educators adopt roles that involve being disciplinarians, empathizers, supporters, and interveners in order to mitigate bullying. But in order for these responsibilities to be carried out successfully, educators must get ongoing training and assistance on bullying prevention. Teachers can be more effective in preventing bullying by using a combination of social-emotional learning strategies like empathy and support in addition to punishment. The study’s conclusions are consistent with research in the literature that highlights the value of peer support for victims of bullying. Positive peer approaches can support the psychological and social recovery processes of bullied students, while negative approaches may perpetuate the cycle of bullying and worsen the victims’ situation. Thus, encouraging empathy, collaboration, and healthful social interactions in schools can help to prevent bullying.

The findings of this study align with previous empirical research that underscores the significance of teachers’ empathetic and supportive roles in both preventing and addressing bullying ([38]; [76]). Additionally, research highlighting peer support as a protective factor against social exclusion ([12]; [72]) supports the observed emphasis on friendship, empathy, and collective intervention among students. This study further contributes to the literature by examining these roles within a dialogic and film-based learning environment, where emotional reflection and moral reasoning are developed collaboratively.

Film-based learning, as evidenced in this study through Wonder, can act as a catalyst for emotional arousal and prosocial modelling, enabling students to engage empathetically with complex social issues. Previous studies support the view that film narratives stimulate moral reasoning and emotional awareness (e.g., [79]; [14]). Nevertheless, scholars have cautioned that films may also oversimplify social conflicts or evoke emotional responses that are not critically mediated ([40]; [64]). Therefore, it is essential that educators guide students in reflective dialogue that contextualizes these emotions and prevents the uncritical internalization of cinematic stereotypes.

Despite its contributions, this study has some limitations. First of all, the study is based solely on student views; the perspectives of other stakeholders such as teachers, parents, or school administration were not considered. Additionally, since the participants’ comments were inspired by a particular movie (Wonder), it can be said that the students responded based on fiction rather than real-life experiences. This shows that the data may be based on perception rather than direct experience. The results also show how students responded to a particular narrative setting because the data collection and classroom discussion were centred around a single film, Stephen Chbosky’s Wonder. This in-depth engagement enhances the likelihood that responses were impacted by the fictitious setting of the film rather than its real-world environment, which restricts the results’ generalisability even though it permits rich qualitative interpretations commensurate with the case study approach. In order to determine whether comparable conversation and social-emotional impacts appear across various plots, age groups, or cultural themes, a wider variety of films or multimodal texts must be included.

Finally, because the study had limited sample size and diversity, the results cannot be generalized to the larger student population. Although the results demonstrate the educational potential of film-based dialogic practices, their implications for teacher preparation and school policy should be viewed as exploratory rather than predictive. The recommendations made here are contextual insights rather than generalizable findings because the data were gathered from a particular school during a brief qualitative intervention. According to the study, incorporating social-emotional learning (SEL) and dialogic pedagogy into class discussions can promote inclusivity, empathy, and introspection. Teachers and teacher educators can use strategies like role models to encourage emotional literacy and open communication in the classroom, but further study is required to see whether these strategies can be applied in other settings and educational systems. The results of this study should be seen as preliminary findings that pave the way for more extensive empirical investigation. Rather than conclusive patterns of effect, insights from the study carried out at a Northern Cyprus primary school point to possible processes. Future studies with several schools, long-term interventions, and a wide range of participants may shed light on the potential effects of film-based dialogic pedagogy on teacher preparation and policy in different settings. Therefore, it is recommended to collect data from different age groups, socio-cultural backgrounds and stakeholders in future research.

### 6.1. Limitations of the Study

When interpreting the results of this study, it is important to take into account a number of limitations.

It has only one shareholder in mind: The ideas and perceptions of kids were the only sources of data; neither the opinions of teachers, parents, or school administrators were taken into account. As a result, interpretations of dialogue and social and emotional learning (SEL) processes primarily take into account the viewpoint of the learner and neglect the ways in which peers or adults might facilitate these encounters.

Reliance on just one movie: Wonder, a film by Stephen Chbosky, was the main subject of data collecting. This restricts generalisation and increases the likelihood that a fictitious setting, rather than actual experiences, shaped students’ moral and emotional reactions. Future studies should investigate whether comparable outcomes occur when several videos or multimodal texts are utilised, even though this single-case approach is in line with a qualitative case study design.

Restricted demographic information: Although the study details the grade levels of the pupils, it is devoid of detailed data regarding the ages, socioeconomic origins, and linguistic diversity of the participants. It is challenging to evaluate the sample’s representativeness or heterogeneity or investigate possible variance among subgroups in the absence of these data.

Restricted context for sampling: One Northern Cyprus primary school served as the source of the participants. Because student replies and conversations may have been impacted by instructor practices, classroom structure, and school culture, this limited context lessens the findings’ transferability. Future investigations would be more robust if they were replicated in various educational institutions and cultural contexts.

### 6.2. Implications for Practice

Despite these limitations, the study offers valuable insights for educators and school administrators. Film-based dialogue pedagogy can be integrated into SEL curricula to create emotionally safe spaces where students can express their feelings, develop empathy, and critically examine social issues such as bullying and inclusivity. Teachers can adapt the dialogue stages used in this study (preview exploration, guided reflection, and collaborative meaning-making) to encourage perspective-taking and moral reasoning. Professional development programs should prepare educators to facilitate open and inclusive dialogue and connect cinematic narratives to SEL competencies. Additionally, school counsellors and administrators can utilize similar film-based activities to foster respectful classroom climates and community values.

### 6.3. Implications for Future Research

Teachers, parents, and administrators should all be included in future studies to increase stakeholder engagement and collect a range of viewpoints on students’ social-emotional growth. It is possible to determine whether dialogic and social interaction outcomes are constant across media kinds by employing comparative designs with various films, genres, or digital tales. Researchers will be able to investigate how socioeconomic position, language background, age, and gender influence dialogic learning participation by gathering more comprehensive demographic data. The findings will be more transferable if the sample is expanded to include different schools or international environments. It will also make it clearer how film-based dialogic pedagogy fosters inclusivity, empathy, and critical thinking in a range of educational circumstances.

## Figures and Tables

**Table 1 behavsci-15-01701-t001:** Participants of the Study.

Gender	Girl: 18	Boy: 17			
Class	4th class: 17	5th class: 18			
Nationality	Cypriot: 15	Turkish: 10	Kurdish: 5	Russian: 3	Arabic: 2
Migrant status	Not migrant: 15	Migrant: 20			
Special needs status	No: 31	Yes: 4			

**Table 2 behavsci-15-01701-t002:** Application Process and Data Collection.

Stage	Data Collection Tool	Methods-Techniques of Application	Dialogic Pedagogy Application	Social Emotional Learning Skills
1st Day				
1. Pre-screening Activity	Unstructured Observation Form	Discussion over movie title and poster	Prior knowledge activation, construction of meaning	Self-awareness, Social awareness
2. Film Screening	Unstructured Observation Form	Free observation of researchers (environment, emotional-behavioural reactions)	Student participation, body language in controlled environment	Emotional awareness, Self-management
2nd Day				
3. Individual Interview	Semi-structured interview questions (main questions)	One-on-one discussion based on interpretation and critical thinking	Individual thought development, open dialogue environment	Self-awareness, Responsible decision making
4. Focus Group Interview	Semi-structured interview questions (sub-questions)	Group work and discussion	Participatory learning, multiple perspectives, collaboration	Relationship building skills, Social awareness
3rd Day				
5. Written Documents	Written open ended interview questions	Written reflection	Reflective thinking, individual expression	Self-awareness, Responsible decision making
2nd and 3rd Day				
6. Class Activities	Activitiy based questions	Letter writing, poster preparation, role playing	Creative expression, experience-based learning, social interaction	Self-management, Empathy, Social awareness, Relationship building

**Table 3 behavsci-15-01701-t003:** Participants’ Perspectives and Emotional Reactions to Responses toward a Bullied Student.

Themes	Codes	Excerpts from Student Opinions
Sadness	Feeling bad	*I would feel bad and sad. I would go to another school.* (S2)
	Crying	*“I would be very upset. I would feel bad. I would cry.”.* (S11)
	Feeling alone	*“I would be upset and feel alone. I would ask them not to make fun of me.”.* (S8)
	Enduring sadness	*I would be very upset because if my friends made fun of me, it would always remain a wound in my heart.* (S10)
Inner struggle	Decrease in self-confidence	*My self-confidence would decrease. I wouldn’t give up and would try to make more friends. In this way, I would have friends who love me for who I am.* (S4)
	Solution seeking	*“I would be very upset because I wouldn’t want to be made fun of for my appearance. I would talk to my teacher and work on resolving the issue.”.* (S5)
Extroverted response	Telling the teacher	*“I would feel like Auggie. If I were made fun of at school, I would tell my classroom teacher.”.* (S6)
	Confronting the mockers	*“I would say, ‘If your face were like mine and everyone made fun of you, would you like it?’“.* (S25)
Seeking connection and support	Family support	*“I would be upset and hurt, but my family would always support me, just like Auggie’s family.”.* (S12)

**Table 4 behavsci-15-01701-t004:** Students’ Views on Teachers’ Roles and Responsibilities in Addressing School Bullying.

Themes	Excerpts from Student Opinions
Disciplining	*“Teacher should warn them. If they continue, the teacher should give a punishment and should extend the punishment until they stop.”* (S33)
	*“Teacher should get angry and make them apologize.”* (S2)
	*“He should say that this is bad behavior and send the child who is acting negatively to the principal.”* (S13)
	*“He should forbid making fun of others.”* (S20)
Showing empathy	*“Teachers should treat him well. They should try to understand him and establish empathy.”* (S5)
	*“Teacher should tell his friends that he is a human too and that he has feelings.”* (S4)
	*“The teacher should ask the other students, what if you were in his place”* (S12)
Providing support	*“Each student should be properly informed. They should be told to get used to Auggie and to respect him. The teacher should also be careful when talking to him.”* (S14)
	*“The teacher should spend time with him.”* (S19)
Intervening and preventing	*“The teacher should protect him. He should prevent him from being bullied.”* (S16)
	*“Assigning a project related to bullying and facilitating discussions among students.”* (S3)

**Table 5 behavsci-15-01701-t005:** Students’ Perspectives on Peer Reactions to Bullying Victims.

Main Theme	Theme	Codes	Excerpts from Student Opinions
Positive Approaches of Peers	Building friendship	Good behaviour	*“They behave well. For example, they can have a meal together.”* (S15)
		Spending time together	*“They go to the park and have fun with him.”* (S18)
		Playing games	*“They play games with him.”* (S26)
	Support	Protection	*“They protect. They can say don’t do it.”* (S7)
		Morale building	*“They can be friends and travel together. They try to fulfil his dreams.”* (S25)
		Empathise	*“What would you do if they treated you like that?”* (Q9)
		Warning	*“They can warn those who mock.”* (S33)
	Finding help	Teacher assistance	*“They can call the teacher and ask for help.”* (S7)
Negative Approach of Peers	Punishment	Resorting to violence	*“They can hit those who exclude him.”* (S27)
	Bullying	Exclusion	*“They may not even look at your face.”* (S24)

**Table 6 behavsci-15-01701-t006:** Students’ Perspectives on Family Responses to Bullying Victims.

Main Theme	Theme	Codes	Excerpts from Student Opinions
Positive Approach of Families	Love and Support	Good behaviour	*“They should try to treat him very well.”* (S33)
		Trust	*“What I like is that parents trust and believe in their children.”* (S24)
		Support	*“I like that the family always supports me”* (S4)
	Empathy and Understanding	Understanding	*“It was nice that they made him feel like a special child and showed understanding”* (S11)
		Empathy	*“I would be the family that empathised more with Auggie.”* (S8)
Negative Approach of Families	Repressive and Unstable	Repressive	*“What I don’t like is the oppressive behaviour of the family.”* (S5)
		Unstable	*“Sometimes they show love, but they do not respect their children’s decisions.”* (S14)
Mixed Feelings of Families	Well-intentioned but lack of understanding	Well-intentioned but lack of understanding	*“I think that even though some families treat their children well, they do not fully understand them.”* (S8)

## Data Availability

The original contributions presented in this study are included in the article/Appendix A. Further inquiries can be directed to the corresponding author.

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
