# Peer review of "Exploring Elementary Students’ Social-Emotional Development Through Dialogic Pedagogy: Insights from Cinematic Narratives"

_behavsci, 2025, doi:10.3390/bs15121701_

Round 1
Reviewer 1 Report
Comments and Suggestions for Authors
Dear authors, I appreciate the opportunity to review your work. I found it very interesting. It was an opportunity to see the film Wonder.
I think the work could be improved if they used a more direct style of expression. It took me several readings to extract the essential elements of the work.
Below, and as an example, the essential elements of the work have been extracted in case they can be used as a guide in the modification proposal.

Author Response
Comment 1: Clarify research questions
Response 2: The research questions are clarified. Each question now specifies focus of inquiry.
Comment 2: Identify basic sociodemographic variables (age and gender distribution)
Response 2: Table 1 is added.
Comment 3: Analysis Categories (IDENTIFY IF ANY ANALYSIS SOFTWARE AND ANY PROCEDURE FOR MEASURING INTERCODER AGREEMENT HAS BEEN USED, E.G., Krippendorf Alpha )
Response 3: The procedure of data analysis is added and explained in detail.
Reviewer 2 Report
Comments and Suggestions for Authors
This manuscript advances an ambitious discourse on “SEL and anti-bullying through dialogic pedagogy.” Bullying is a salient issue in children’s lives and a worthy topic of inquiry; however, the study’s execution presents substantial limitations.
Major
Introduction
Although the manuscript cites relevant theories and prior research, the research gap (so-what) and the research questions are not articulated with sufficient clarity. While it names key terms and the local context (the multicultural situation in Cyprus), the conceptual bridge to the study design remains tenuous.
(1) Specify two to three concrete gaps in prior work (e.g., affect-induction bias when using film; lack of dialogue transcripts as primary data), and
(2) redefine RQ1–RQ3 in terms of measurable analytic units (emotion, behavior, social interaction).
Literature Review
The conceptual boundaries and operating mechanisms among SEL, dialogic pedagogy, and film-based learning are presented too abstractly (i.e., which interactional moves are expected to affect which SEL subdomains). Arguments concerning teacher roles and peer support that appear in the results/discussion are recycled in the conclusion without rigorously checking alignment with prior empirical studies. Place supportive and critical evidence regarding film as a driver of emotional arousal and social modelingside-by-side to provide a balanced review.
Validity of the Research Design
In practice, the dataset is a single-case study collecting responses after the screening of a single film (Wonder), rendering the scope of data very limited. The empirical support is thin relative to the scale of the theoretical claims. A one-shot stimulus–response design centered on immediate post-screening reactions is insufficient for claims that “dialogic pedagogy enhances SEL.” At minimum, include pre–post data (teachers, parents, and classroom observations) or longitudinal classroom tracking. Alternatively, implement a multi-session program (e.g., 5–6 films with dialogic sessions). The current study over-relies on self-reports and students’ written narratives.
Concrete requirements:
(1) Provide Table 1 detailing participant characteristics, recruitment route, and attrition;
(2) Diagram a timeline of data collection before/during/after class activities (screening, discussion, worksheets);
(3) Describe how researchers facilitated/mediated dialogic practices with specific procedures.
Methods
Reporting on coding procedures, reliability, saturation criteria, and exemplar quotes is seriously insufficient. Although “content analysis” is mentioned, unit of analysis, codebook excerpts, and inter-coder agreement—pillars of qualitative rigor—are missing. Absent codebook, illustrative quotations, saturation criteria, and reliability indices (e.g., Cohen’s κ / percent agreement) markedly undermines replicability and auditability.
Sampling
Although the study states that 35 participants were purposively sampled, it provides no analysis or description of grade level, gender, migration background, or special educational needs—factors that substantially shape bullying experiences and emotional responses. Basic information about the sample must be presented. Include sampling criteria and a participant characteristics table (grade/gender/background/recruitment path), etc.
Results
While summary tables (Tables 2–5) are offered, participants’ direct/indirect quotations (1–3 sentences) are too sparse or partial. The manuscript requires more raw data “voice” to substantiate claims and interpretations. Reorganize the section so that evidence → interpretation → theoretical linkage is explicit and traceable.
Discussion
The manuscript over-extends policy and teacher-education implications from single-school, short-term qualitative data. The strength of claims should be scaled back. Frame recommendations for school policy and teacher education as exploratory and place the contextual constraints of this study front and center.
Minor
Editing & Language: Address English typos, spacing, and redundancies; ensure terminological consistency (names of SEL components; publication years). Improve logical flow in the abstract.
Reference Accuracy: Verify year/spelling consistency between in-text citations and references. Reduce unsupported generalizations.
Author Response
- The research gap (so-what) and the research questions are not articulated with sufficient clarity. While it names key terms and the local context (the multicultural situation in Cyprus), the conceptual bridge to the study design remains tenuous.
Specify two to three concrete gaps in prior work (e.g., affect-induction bias when using film; lack of dialogue transcripts as primary data)
Located as a paragraph in the introduction part after the paragraph starting with “Bullying is such an important…”
- Redefine RQ1–RQ3 in terms of measurable analytic units (emotion, behavior, social interaction).
The research questions are clarified. Each question now specifies focus of inquiry.
- The conceptual boundaries and operating mechanisms among SEL, dialogic pedagogy, and film-based learning are presented too abstractly (i.e., which interactional moves are expected to affect which SEL subdomains).
Located at the end of theoretical framework part just before methodology
- Arguments concerning teacher roles and peer support that appear in the results/discussion are recycled in the conclusion without rigorously checking alignment with prior empirical studies. Place supportive and critical evidence regarding film as a driver of emotional arousal and social modelingside-by-side to provide a balanced review.
Two new paragraphs are added to the conclusion and recommendations part right after the sentence starting with “thus encouraging empathy, collaboration…”
- In practice, the dataset is a single-case study collecting responses after the screening of a single film (Wonder), rendering the scope of data very limited. The empirical support is thin relative to the scale of the theoretical claims. A one-shot stimulus–response design centered on immediate post-screening reactions is insufficient for claims that “dialogic pedagogy enhances SEL.” At minimum, include pre–post data (teachers, parents, and classroom observations) or longitudinal classroom tracking. Alternatively, implement a multi-session program (e.g., 5–6 films with dialogic sessions). The current study over-relies on self-reports and students’ written narratives.
Unfortunately, it is not possible to make this suggestion. This is because the research would need to be redesigned. To address the referee's concerns, a long and detailed limitations section has been written in the results section.
- Provide Table 1 detailing participant characteristics, recruitment route, and attrition. Although the study states that 35 participants were purposively sampled, it provides no analysis or description of grade level, gender, migration background, or special educational needs—factors that substantially shape bullying experiences and emotional responses. Basic information about the sample must be presented. Include sampling criteria and a participant characteristics table (grade/gender/background/recruitment path), etc.
Table 1 is added.
- Diagram a timeline of data collection before/during/after class activities (screening, discussion, worksheets)
In Table 2, has already been indicated as 1st day, 2nd day etc.
- Describe how researchers facilitated/mediated dialogic practices with specific procedures.
A new paragraph as well as the indication of the process days were added to the process section in method. By doing so we illustrated how dialogic pedagogy was operationalized in classroom practice.
- Reporting on coding procedures, reliability, saturation criteria, and exemplar quotes is seriously insufficient. Although “content analysis” is mentioned, unit of analysis, codebook excerpts, and inter-coder agreement—pillars of qualitative rigor—are missing. Absent codebook, illustrative quotations, saturation criteria, and reliability indices (e.g., Cohen’s κ / percent agreement) markedly undermines replicability and auditability.
Suggested info is added to method section and as a supplementary document.
- While summary tables (Tables 2–5) are offered, participants’ direct/indirect quotations (1–3 sentences) are too sparse or partial. The manuscript requires more raw data “voice” to substantiate claims and interpretations. Reorganize the section so that evidence → interpretation → theoretical linkage is explicit and traceable.
Recommended change done. One paragraph is added after each table in findings part.
- The manuscript over-extends policy and teacher-education implications from single-school, short-term qualitative data. The strength of claims should be scaled back. Frame recommendations for school policy and teacher education as exploratory and place the contextual constraints of this study front and center.
The policy and teacher-education implications have been reframed as exploratory insights rather than prescriptive recommendations.
- Address English typos, spacing, and redundancies; ensure terminological consistency (names of SEL components; publication years).
Checked and corrected.
- Improve logical flow in the abstract.
Abstract was improved and logical flow enhanced.
- Verify year/spelling consistency between in-text citations and references.
Verified and corrected.
- Reduce unsupported generalizations.
Done. The language and wording controlled according to the reviewer suggestion
Reviewer 3 Report
Comments and Suggestions for Authors
Thank you for the opportunity to review the manuscript titles, Exploring Elementary Students' Social-Emotional Development Through Dialogic Pedagogy: Insights from Cinematic Narratives. This study addresses an important issue in education, i.e., how dialogic pedagogy and film-based learning can be used to support social-emotional learning (SEL) and address bullying in diverse classroom contexts. Below, I provide section-by-section feedback to highlight the manuscript’s strengths as well as areas where it could be improved.
Introduction: Introduction situates the study within the broader literature on dialogic pedagogy and SEL. It provides justification for why the Cypriot context is unique and significant especially given issues of migration and bullying. Connection between bullying prevention and dialogic pedagogy is supported with relevant citations and research. The manuscript could be strengthened by the following revisions:
- Justification for choosing film (Wonder) as the pedagogical tool is not fully developed in this section.
- Research Case and Questions subsection currently sits at the start of the Methods, but it would be more logically placed at the end of the Introduction.
Method: Method section shows good effort in employing multiple qualitative data collection strategies, including observations, focus groups, individual interviews, and written reflections. Triangulation of data sources is a strength. Ethical approval and consent are also noted. The manuscript could be strengthened by the following revisions:
- Research Case and Questions subsection currently sits at the start of the Methods, but it would be more logically placed at the end of the Introduction.
- Age range, gender distribution, and other demographic information of the 35 participants are not provided.
- Choice of purposive sampling is described as based on accessibility and convenience. This undermines claims of contribution to underexplored contexts. More explanation is needed regarding why this school and group were appropriate for the research questions.
- Authors note that two researchers facilitated and observed but provide no information about their expertise or training in conducting interviews with children, managing group discussions, or handling sensitive topics such as bullying.
- While open-ended questions are mentioned, there is no description of how the researchers ensured that children felt safe, comfortable, and able to express themselves authentically. For example, strategies to minimize power imbalance, establish rapport, or adapt questioning to developmental levels are not described.
- Content analysis process is outlined in general terms, but there is little detail about coding procedures, use of software (if any), or how disagreements between coders were resolved beyond consensus. Reliability checks (e.g., inter-coder agreement, peer debriefing, or audit trails) are missing.
- Common qualitative measures such as member checking, thick description, or triangulation of researcher perspectives are not described sufficiently.
- While the rationale for selecting this film is touched on, the narrow focus raises questions about the robustness of findings. A stronger case could be made for why this particular narrative is pedagogically significant beyond its popularity. Heavy reliance on a single film (Wonder) limits generalizability.
Findings: Findings are organized according to the research questions, and the themes are supported with illustrative student quotes. The manuscript could be strengthened by the following revisions:
- The results are descriptive; they mainly report participant responses and list themes without sufficient interpretive analysis.
- Heavy reliance on tables and direct quotes means that the findings read more like raw data presentation rather than synthesized insights.
- Connections to theoretical constructs (e.g., SEL competencies, dialogic pedagogy principles) are missing which is leaving the results section disconnected from the conceptual framework.
- There is little to no discussion of patterns across subgroups (e.g., age, gender, different experiences of bullying). Because demographic details are missing in the methods, the findings cannot be meaningfully contextualized.
- The results section does not clearly demonstrate how the analytic process (coding, theme development, consensus) shaped the reported themes.
Discussion: Discussion highlights the importance of empathy, peer support, and teacher intervention in addressing bullying, aligning the findings with existing SEL and dialogic pedagogy literature. It emphasizes the value of dialogic classroom practices for fostering inclusive and supportive environments. The manuscript could be strengthened by the following revisions:
- Limitations of the study, implications for practice, and implications for research should be expanded and presented under separate subsections.
- Limitations should include the following at minimum:
- study relies solely on student perceptions, excluding other stakeholders such as teachers, parents, or administrators.
- Data collection is tied to one specific film (Wonder), which restricts generalizability and raises concerns that findings reflect students’ reactions to fictional narratives rather than real-life experiences.
- Demographic details (age range, gender, socio-economic background) are missing, making it difficult to assess how representative or diverse the participants were.
- sample is limited to one school, reducing the transferability of findings
Overall, this manuscript useful perspective into the potential of dialogic pedagogy and film-based learning to support SEL and address bullying in culturally diverse classrooms. However, the paper would benefit from stronger methodological justification, and in-depth theoretical synthesis in the results and discussion.
Author Response
- Justification for choosing film (Wonder) as the pedagogical tool is not fully developed in this section.
Justification for choosing the film was developed and located in section 3.5-
“Data collection tools”
- Research Case and Questions subsection currently sits at the start of the Methods, but it would be more logically placed at the end of the Introduction.
It is placed at the end of the introduction.
- Age range, gender distribution, and other demographic information of the 35 participants are not provided.
Table 1 is created and socio-demographic info is added.
- Choice of purposive sampling is described as based on accessibility and convenience. This undermines claims of contribution to underexplored contexts. More explanation is needed regarding why this school and group were appropriate for the research questions.
Choice of purposive sampling is described based on accessibility and convenience (See 3.2)
- Authors note that two researchers facilitated and observed but provide no information about their expertise or training in conducting interviews with children, managing group discussions, or handling sensitive topics such as bullying.
Detailed information is added about the researchers of the study (See 3.3.)
- While open-ended questions are mentioned, there is no description of how the researchers ensured that children felt safe, comfortable, and able to express themselves authentically. For example, strategies to minimize power imbalance, establish rapport, or adapt questioning to developmental levels are not described.
The suggested information is added and explained in the 'Process' section of the method.
- Content analysis process is outlined in general terms, but there is little detail about coding procedures, use of software (if any), or how disagreements between coders were resolved beyond consensus. Reliability checks (e.g., inter-coder agreement, peer debriefing, or audit trails) are missing.
All the information is provided and explained in the 'Data Analysis' section of the 'Methods'.
- Common qualitative measures such as member checking, thick description, or triangulation of researcher perspectives are not described sufficiently.
Some more explanations have been added to provide sufficient description.
- While the rationale for selecting this film is touched on, the narrow focus raises questions about the robustness of findings. A stronger case could be made for why this particular narrative is pedagogically significant beyond its popularity. Heavy reliance on a single film (Wonder) limits generalizability.
Wider explanation added as required.
- The results are descriptive; they mainly report participant responses and list themes without sufficient interpretive analysis.
Interpretive analysis is done for each table.
- Heavy reliance on tables and direct quotes means that the findings read more like raw data presentation rather than synthesized insights.
As we indicated for reviewer’s tenth recommendation, one additional paragraph is added for each table
- Connections to theoretical constructs (e.g., SEL competencies, dialogic pedagogy principles) are missing which is leaving the results section disconnected from the conceptual framework.
Recommended connection to theoretical construct is added to the last paragraph of theoretical framework section. And the study’s results connected and explained in detail in conclusion.
- There is little to no discussion of patterns across subgroups (e.g., age, gender, different experiences of bullying). Because demographic details are missing in the methods, the findings cannot be meaningfully contextualized.
Suggested discussion is added to the last paragraph of the discussion section in order to give more demographic details.
- The results section does not clearly demonstrate how the analytic process (coding, theme development, consensus) shaped the reported themes.
Coding source and analysis process are given in Supplementary documents.
- Limitations of the study, implications for practice, and implications for research should be expanded and presented under separate subsections. Limitations should include the following at minimum:
- study relies solely on student perceptions, excluding other stakeholders such as teachers, parents, or administrators.
- Data collection is tied to one specific film (Wonder), which restricts generalizability and raises concerns that findings reflect students’ reactions to fictional narratives rather than real-life experiences.
- Demographic details (age range, gender, socio-economic background) are missing, making it difficult to assess how representative or diverse the participants were.
- sample is limited to one school, reducing the transferability of findings
A new section titled “6. Limitations and Implications” has been added. It now includes three clearly defined subsections—6.1 Limitations of the Study, 6.2 Implications for Practice, and 6.3 Implications for Future Research. The revised text explicitly addresses all points noted by the reviewer: reliance on student perceptions, use of a single film (Wonder), limited demographic details, and a single-school sample. The implications have been expanded and reframed as exploratory to reflect the contextual constraints of this qualitative case study and to guide future research directions.
Round 2
Reviewer 3 Report
Comments and Suggestions for Authors
Thank you for the opportunity to review the revised version of the manuscript titled Exploring Elementary Students' Social-Emotional Development Through Dialogic Pedagogy: Insights from Cinematic Narratives. I would like to acknowledge the authors for carefully addressing all of the concerns raised during the previous review cycle. The revisions considerably strengthened the paper’s conceptual alignment, methods and analytic depth.
Overall, the revised manuscript provides a meaningful contribution to the literature on social-emotional learning and dialogic pedagogy in culturally diverse elementary settings.